# Investigating the Prevalence of RNA-Binding Metabolic Enzymes in *E. coli*

**DOI:** 10.3390/ijms241411536

**Published:** 2023-07-16

**Authors:** Thomas Klein, Franziska Funke, Oliver Rossbach, Gerhard Lehmann, Michael Vockenhuber, Jan Medenbach, Beatrix Suess, Gunter Meister, Patrick Babinger

**Affiliations:** 1Institute of Biophysics and Physical Biochemistry, Regensburg Center for Biochemistry, University of Regensburg, D-93040 Regensburg, Germany; 2Institute of Biochemistry, Faculty of Biology and Chemistry, University of Giessen, D-35392 Giessen, Germany; 3Institute of Biochemistry, Genetics and Microbiology, Regensburg Center for Biochemistry, University of Regensburg, D-93040 Regensburg, Germany; 4Centre for Synthetic Biology, Technical University of Darmstadt, D-64287 Darmstadt, Germany

**Keywords:** metabolic enzymes, REM hypothesis, RNA-binding protein, prokaryotes, SELEX, unconventional RNA binding, MS2 phage, quinone oxidoreductase

## Abstract

An open research field in cellular regulation is the assumed crosstalk between RNAs, metabolic enzymes, and metabolites, also known as the REM hypothesis. High-throughput assays have produced extensive interactome data with metabolic enzymes frequently found as hits, but only a few examples have been biochemically validated, with deficits especially in prokaryotes. Therefore, we rationally selected nineteen *Escherichia coli* enzymes from such datasets and examined their ability to bind RNAs using two complementary methods, iCLIP and SELEX. Found interactions were validated by EMSA and other methods. For most of the candidates, we observed no RNA binding (12/19) or a rather unspecific binding (5/19). Two of the candidates, namely glutamate-5-kinase (ProB) and quinone oxidoreductase (QorA), displayed specific and previously unknown binding to distinct RNAs. We concentrated on the interaction of QorA to the mRNA of *yffO*, a grounded prophage gene, which could be validated by EMSA and MST. Because the physiological function of both partners is not known, the biological relevance of this interaction remains elusive. Furthermore, we found novel RNA targets for the MS2 phage coat protein that served us as control. Our results indicate that RNA binding of metabolic enzymes in procaryotes is less frequent than suggested by the results of high-throughput studies, but does occur.

## 1. Introduction

Massive efforts are being expended to understand the complex interactome within living organisms. Protein–RNA interactions are an essential component and contribute to cellular regulation in manifold ways. The traditional view of an RNA-binding protein (RBP) is governed by clearly defined RNA-binding domains that the protein uses to influence the fate of specific RNAs, thereby exerting regulation. RBPs can combine several such domains to expand specificity [1]. Proteins harboring well-defined RNA-binding domains are often termed “canonical RBPs” in the literature. However, it is becoming increasingly evident that the interactome is not exclusively defined by canonical RNA binding. Instead, there are various modes of “unconventional RNA binding”. For example, RNAs can function as a scaffold to guide inter-chromosomal interactions in eukaryotes [2], act on architectural proteins to impact the organization of chromatin [3] or guide the agglomeration of glycolytic enzymes in yeast [4]. Such case examples support the notion that not only can proteins guide RNA function, but also the other way around. Intrinsically disordered regions are being recognized as important mediators of RNA binding [5] and have opened up a new field of RNA-binding research. Disordered regions occur in various contexts: They are found as linkers between RNA-binding domains [6], and they were shown to engage in highly specialized functions like phase separation of ribonucleoprotein granules [7].

The RNA–enzyme–metabolite (REM) hypothesis, an acronym introduced more than a decade ago [8], suggests regulatory interplay of RNAs, metabolic enzymes and metabolites. This concept proposes functional complexity for enzymes that goes beyond their well-understood catalytic properties. Over time, particular attention to such unconventional RNA binding has been drawn by high-throughput methods used to identify RBPs. Screenings utilizing micro arrays and mass spectrometry collectively propose RNA-binding activities for dozens of metabolic enzymes, but until a few years ago, such studies were feasible in eukaryotes only [9,10,11,12].

While supported by a few reports in the literature, the prevalence of a REM network is currently unclear, especially in prokaryotes. The most prominent example is eukaryotic aconitase IRP1. This enzyme participates in the citric acid cycle by converting citrate to isocitrate. However, the enzymatic activity depends on an iron-sulfur cluster that is depleted under iron-starvation conditions. In the absence of the metallocofactor, the protein engages in posttranscriptional regulation and interacts with mRNAs that impact iron homeostasis [13,14]. Another enzyme with reported RNA-binding functions is glyceraldehyde-3-phosphate dehydrogenase. Multiple specific interactions have been reported, including binding with tRNA in HeLa cells [15], AU-rich elements of human IFN-γ 3′-UTR mRNA regions [16], human GLUT1 3′-UTR mRNA [17], hepatitis A virus RNA [18], and parainfluenza virus type 3 RNA [19]. Notably, the exact site and mode of binding have not been conclusively elucidated until recently—the Rossmann-fold domain, substrate-binding groove, dimerization interface, and an RRM-like structural motif have all been mentioned to be potentially involved in RNA binding [20,21,22,23]. Other metabolic enzymes that have been reported to exert RNA-binding activities are: human thymidylate synthase, [24], human dihydrofolate reductase [25], flesh fly thiolase [26], bovine glutamate dehydrogenase [27], yeast mitochondrial isocitrate dehydrogenase [28], human glucose-6-phosphate dehydrogenase [17], human lactate dehydrogenase [29], mouse aldolase [30], and human phosphoglycerate kinase [31]. Thorough characterizations of these proposed interactions are missing for almost all cases until recently. The example of iron homeostasis by aconitase arguably represents the only precisely understood “REM interaction” in the sense of mutual regulation between the three players RNA, enzyme, and metabolite. The prevalence in cells and differences across domains of life remain key questions to be answered.

In recent years, high-throughput screening methods that facilitate unbiased RBP identification in bacteria have been developed and applied [32,33,34]. Like in datasets from eukaryotes, many metabolic enzymes have been identified as hits, suggesting that dozens of potential REM interactions are yet to be discovered. We set out to obtain more perspective on this suggested frequent occurrence in prokaryotes, for which protein-centered investigations are particularly scarce. For this purpose, we probed various *E. coli* enzymes for specific RNA-binding properties that might hint at physiologically relevant REM functions. Our investigations were mainly based on SELEX (**s**ystematic **e**volution of **l**igands by **ex**ponential Enrichment) experiments, probing for preferential binding behavior within a library representing the transcriptome of E. coli, but also included iCLIP (**i**ndividual-nucleotide resolution UV **c**ross**l**inking and **i**mmuno**p**recipitation) and EMSA (**e**lectrophoretic **m**obility **s**hift **a**ssay) studies.

## 2. Results

### 2.1. Selection of Proteins and Strategy

We picked a set of nineteen metabolic enzymes from *E. coli* as potential RNA-interacting proteins. The selection was based on recurring literature references to putative RNA binding of metabolic enzymes in both high-throughput screenings and targeted research. We included data for non-prokaryotic enzymes that are functionally and structurally homologous. A particular set of recently published interactome screenings in prokaryotes was given special consideration. These screenings used phase separation [32,33] or silica bead purification [34] as unbiased methods to extract crosslinked RNA–protein complexes from bacterial cells. Considering that polyadenylation in bacteria is much less abundant compared to that in eukaryotes and that the widely used poly(A) capturing cannot be used for holistic RNA isolation, such alternative purification methods bring new insights not only into the bacterial mRNA interactome, but into RNA interactomes in general. All candidates selected by us were significant hits in at least one of these screens (Appendix A). Table 1 provides a list of all target proteins, sorted by metabolic pathway, as well as references to additional literature describing associated RNA-binding features. Some representatives are frequently mentioned in the literature as suspected or even confirmed RNA-binding proteins (with GAPDH being an example of the latter). Trying to understand the general prevalence of REM interactions in bacteria, we chose candidates from a wide range of metabolic pathways.

We applied iCLIP as well as protein SELEX to identify interactions between *E. coli* RNA species and the selected enzymes (Figure 1). For iCLIP, we modified respective *E. coli* genes with FLAG-tags to enable immunoprecipitation of in vivo-crosslinked RNA–enzyme complexes. For SELEX, we employed an *E. coli* genomic DNA fragment pool, transcribed it into an RNA library, and exposed it to each of the enzymes during iterative selection rounds.

### 2.2. iCLIP for Selected Enzyme Candidates Gives No Hints on Specific RNA Binding

We opted for iCLIP experiments, because they utilize UV crosslinking to capture protein–RNA interactions under in vivo conditions. The application of CLIP experiments, while frequent for the study of eukaryotic proteins, has, for prokaryotes, been hitherto limited to key regulators Hfq, CsrA, ProQ, and the archaeal exosome [44,45,46,47]. iCLIP experiments were carried out as described previously [48], with minor adaptions to account for the bacterial cell. FLAG tags were introduced on the genomic level into the *E. coli* genes of interest to allow immunoprecipitation, using the established methods of λ-red mediated recombination [49,50].

iCLIP experiments were performed with the RBP candidates Mdh, PykF, RpiA, KdsA, Upp, Pgk, and AnsB. The known RNA binders Rho [51] and Hfq [52] served as positive controls. As negative controls, samples of a strain without a FLAG tag and cells not exposed to UV irradiation were processed. While all controls behaved as expected, only weak signals could be obtained for Mdh, Pgk, PykF, Upp, AnsB, and KdsA, but not for RpiA. Based on these results, iCLIP was carried out in a preparative setup for Mdh, Pgk, PykF, Upp, AnsB, and KdsA in replicates. Co-immunoprecipitated RNAs were isolated, reverse-transcribed, processed into a multiplexed library, and subjected to next-generation sequencing. Background controls (size-matched input, SMI [53]) that represent all RNAs bound to proteins of similar size were included.

Inspection of the sequencing results revealed that both RBP candidates and SMI samples featured a significant overrepresentation of tRNA and rRNA sequences (>95% of reads), which are the most abundant RNA species in the cell. Identified clusters among coding and ncRNAs (<5% of reads) coincided with abundant transcripts like, for example, the lipoprotein mRNA (*lpp*). Remarkably, the six different tested enzymes primarily captured the same abundant RNA targets. Overall, this indicates that iCLIP with these non-canonical RBPs captured RNAs based on their generally high abundance and, therefore, their random close proximity to the tested enzymes in the cell rather than a specific interaction. A careful optimization of conditions might help alleviate such unspecific crosslinking of abundant RNAs. Because we performed the SELEX experiments (see Section 2.3) in parallel, and they provided more promising results, we refrained from optimizing the iCLIP experiments and from testing the other RBP candidates. In summary, four of the six candidates that captured RNA unspecifically in iCLIP led to no enrichment of specific sequences in SELEX. The other two (PykF and Pgk) enriched only poorly defined RNA patterns in SELEX. Therefore, the iCLIP results are in good agreement with SELEX results, as iCLIP only enriched the same highly abundant RNAs for all tested candidates. For brevity, we do not provide figures for those results, but the sequencing results are available in a survey as Appendix A and as raw data at http://doi.org/10.5283/epub.54333, accessed on 14 July 2023

### 2.3. Multiple Enzymes Show Enrichment of Specific RNAs in Genomic SELEX Experiments

Each enzyme candidate was exposed to an RNA pool transcribed from an *E. coli* genome-fragment library in at least six iterative cycles. Resulting evolved libraries were subjected to next-generation sequencing. The reads were mapped to the genome of *E. coli* to identify read agglomeration (“clusters”), an indicator for potential specific RNA binding. We tested a total of 18 metabolic enzymes (see Table 1; RpiA was not tested in SELEX), and seven of them showed a principally unique cluster formation. As discussed in detail below, these included pyruvate kinase (PykF), phosphoglycerate kinase (Pgk), thymidylate synthase (ThyA), glutamate-5-kinase (ProB), aconitase (AcnB), glyceraldehyde-3-phosphate dehydrogenase (GapA), and quinone oxidoreductase (QorA). For each of these enzymes, a distribution map of sequenced reads is shown in Figure 2 (lines 3–9). For the remaining enzymes that did not feature unique cluster formation, analogous read distribution maps are provided in Appendix A. We executed two control experiments to ensure the integrity of our SELEX results. The first control (Figure 2, line 1) comprised exclusive, repeated PCR amplification (resembling the number of PCR cycles during actual protein SELEX). It produced an evenly flat distribution of reads throughout the *E. coli* genome, verifying that clusters originating from mere PCR bias do not reach sizes comparable to those from protein selection. As a second level of control experiment, MS2 phage coat protein was employed (Figure 2, line 2). MS2 coat protein is known to specifically bind heterologous *E. coli* RNAs [43,54] and was thus expected to display a strong selection effect in SELEX.

We assume that unique clusters on their own are not a guaranteed indication of high-affinity interactions between the employed protein and respective RNAs. They rather attest to principal discrepancies in affinity of undetermined magnitude. As such, they comprise a meaningful starting point to examine amplified RNAs and probe for binding in follow-up experiments. To identify RNA features that might exert specific binding, we looked for all significant read clusters in the genome mapping and analyzed them via MEME motif inquiry as well as RNA secondary structure prediction of the underlying transcript. We found distinct sequence and structural features within the respective enriched RNAs, which for some enzymes comprised the majority of reads within the whole sample. These features were very different for each candidate protein. In the following paragraphs, the identity of the features is discussed in detail for each enzyme (as well as for control MS2 coat protein), and a summary is given in Table 2. Intriguingly, connections to REM or the existing literature can be drawn in multiple cases. Our sequencing raw data are available at http://doi.org/10.5283/epub.54333, accessed on 14 July 2023.
SELEX results for MS2 phage coat protein


While not representative of *E. coli* metabolism, we still wanted to report SELEX results for our positive control, the MS2 phage coat protein, a well-characterized model RBP for RNA recognition. The primary target of the MS2 coat protein is an operator hairpin in the phage replicase mRNA to repress its translation [55]. Genomic SELEX experiments previously showed that the coat protein also binds to heterologous *E. coli* RNA fragments that resemble this hairpin [54] (consensus sequence shown in Figure 3A). Our experiment yielded the same trend. The seven biggest read clusters comprised ~87% of total mapped reads after six SELEX rounds, and the sequence logo derived from clustered reads represented the operator hairpin consensus sequence, constituting a stem with a pronged adenosine and the four-base loop ANYA (Figure 3A). We found the Y position to be consistently occupied by a cytosine, confirming a reported cytosine-specific affinity increase [56].

Read clusters observed by us included five of nine genes identified previously in the SELEX experiment by Shtatland et al. [54]. A complete list of the genomic loci identified by us is provided in Appendix A. Interestingly, virtually all identified genes carried functions directly or indirectly connected to cell surface. This held true for all prominent read clusters—*dacB* (peptidoglycan synthesis), *rffG* (enterobacterial common antigen biosynthesis), *tesA* (fatty acid metabolism), *sslE* (biofilm maturation), *fliO* (flagellar protein), *alx* (putative membrane-bound redox modulator), and *ecpC* (pilus formation, biofilm), as well as for many other genomic loci listed in Appendix A. Therefore, we interpret our data as further evidence for the involvement of the MS2 phage coat protein in host gene regulation, as was previously hypothesized by Shtatland et al., and we emphasize the need of further investigations on a potential regulatory impact of MS2 coat protein on surface development in *Enterobacteriaceae* cells.

Exemplarily, we validated the binding of the coat protein to the *rffG* RNA in an EMSA experiment (Figure 3B). Binding was observable already at low coat protein concentrations, and an unrelated competitor RNA with random sequence could not disrupt the MS2 coat protein:*rffG* complex.
Pyruvate kinase (PykF)

Initial experiments had shown that pyruvate kinase has an unspecific RNA-binding activity with a value for the dissociation constant (K_d_) in the low micromolar range (Appendix A), and SELEX should elucidate whether there are particular RNAs that show a stronger binding. We identified multiple read clusters in the genome mapping, but we identified no communalities within the underlying RNA species, neither for sequence nor for the predicted structure. Because various unrelated binding motifs were assessed as highly unlikely, we concluded that our SELEX data reaffirm the unspecific RNA-binding property of PykF. We hypothesize that insignificant affinity discrepancies led to a snowballing accumulation of slightly favored RNA sequences during the multiple SELEX rounds. This underlines that diligent interpretation of sequencing data and follow-up experiments are vital to deduce specific binding from iterative selection methods.
Phosphoglycerate kinase (Pgk)


Exposing the genomic RNA library to phosphoglycerate kinase led to a strong enrichment of several sequences that form similar hairpin structures. While not strictly sequence-conserved, we found the loop sequences to be dominated mainly by adenosines, and to a lesser extent by cytosines (Figure 4). While the strong read clustering seemed indicative for specific binding, we observed only weak shifted bands to those sequences in EMSA experiments, but increased affinity to a short, linear A-rich control RNA fragment (Appendix A). Similarly, Pgk showed binding toward an RNA of unspecific sequence (Appendix A). Pgk comprises two Rossmann-fold domains, responsible for ATP binding and substrate binding, respectively [57]. While the Rossmann fold is in fact reported to confer specific RNA binding in certain proteins [16], it is also comprehensible that Pgk selected the hairpins based on a coincidental interaction involving the protein’s nucleotide binding pocket and the A-rich loop, considering the metabolic adenine nucleotide binding function of Pgk. We assume that the hairpin structures were selected based on minor affinity discrepancies, which can still produce enrichment during iterative SELEX rounds. While it cannot be ruled out that our results do in fact suggest physiologically relevant protein–RNA interactions, they should be interpreted with care.Glyceralde-3-phosphate dehydrogenase (GapA)


Eukaryotic glyceralde-3-phosphate dehydrogenase is frequently referred to as a noncanonical, AU-rich element RNA-binding protein [23,58]. Despite many studies affirming its principal RNA-binding function, the identity of the binding site and the scope of biologically relevant RNA targets had remained elusive until recently. Evaluation of read clusters in our SELEX experiment showed a moderate enrichment of AU-rich stretches, albeit with no conserved nucleotide sequence (Table 2). AU-rich fragments accounted for around 14% of the total number of mapped reads, and also had the highest individual read numbers. A list of these fragments is provided in Appendix A. Due to our result remotely resembling the various reports of AU-rich binding for eukaryotic glyceralde-3-phosphate dehydrogenase [16,59,60], we deduced that the bacterial representative also could carry a biologic function as a specific RBP. To the best of our knowledge, this is the first investigation of bacterial glyceralde-3-phosphate dehydrogenase with regard to sequence-specific RNA binding. We did not conduct follow-up binding assays due to the lack of a common sequence motif within enriched RNA fragments, but emphasize that further studies on bacterial glyceraldehyde-3-phosphate dehydrogenase as an AU-rich element-binding protein should be illuminating.
Thymidylate Synthase (ThyA)


The RNA library iteratively exposed to thymidylate synthase featured an unambiguous amplification of a conserved RNA sequence predicted to form a stem-triloop structure (Table 2). A highly significant proportion (>98%) of total mapped reads clustered at genomic loci predicted to carry such stem-loops within transcripts. A sequence motif derived from all read clusters as well as sequences of the seven strongest enriched RNAs are shown in Figure 5 (a comprehensive list of read clusters can be found in Appendix A). Intriguingly, the structure strongly resembles an iron-responsive element found in human amyloid-beta-precursor protein mRNA, which features a hairpin GC(AGA)GC with the same AGA triloop [61]. The hairpin amplified by ThyA, however, carries a strictly conserved non-Watson Crick G-U base pairing within the stem. Notably, our results revealed no enrichment of *thyA* mRNA, which is reportedly bound by its own gene product ThyA at an unknown binding site in *E. coli* [38]. Hence, a particular connection between results on hand and the previously reported feedback interaction could not be drawn. However, stem-loop structures reportedly bound by eukaryotic thymidylate synthase also feature a non-Watson Crick G-U base pair, albeit maintaining larger loops [24,62]. Eukaryotic thymidylate synthase was also shown to select a similar stem-triloop structure, CG(UGU)CG, out of a completely randomized 25 nt RNA library and bind to it with high affinity, featuring relatively high robustness to nucleotide substitutions within the triloop [63]. Surprisingly, we were not yet able to establish a specific interaction between the hairpin and *E. coli* thymidylate synthase in EMSA experiments. Nevertheless, the unambiguously strong enrichment during SELEX made us hypothesize that bacterial thymidylate synthase also might maintain biologic functions by binding to specific hairpin structures, thus calling for further in vitro validation.Aconitase (AcnB)


We tested the *E. coli* homologue of eukaryotic aconitase, a prime example of a dual-functioning, RNA-binding metabolic enzyme. We observed no read amplification occurring at the 3′ UTR site of *acnB*. Published data indicate that inactivated *E. coli* aconitase, i.e., aconitase not occupied by an iron-sulfur cluster, self-regulates its own mRNA stability by binding to 3′ UTR secondary structure elements, albeit differing from iron-responsive elements (IREs) identified as interactors in eukaryotes [64]. Nevertheless, we observed enrichment of specific RNAs. Read clusters were found at the 5′ UTR site of *rplJ*, *copA*, and *yejM*, as well as the 3′ UTR site of *ymgC* and *sbcC*. Transcripts of those genomic loci carry AU-rich stretches with 5′-adjacent hairpins according to secondary structure prediction, as illustrated in Figure 6. Intriguingly, these structures resemble features within the 3′ UTR transcript of *E. coli acnB*, which likewise carries AU-rich stretches with 5′-adjacent hairpins, reported to be involved in AcnB feedback binding [41,64]. A secondary structure prediction of this 3′ UTR *acnB* site is provided in Figure 6 (right side), highlighting the similarity to RNA fragments enriched in our experiment. We did not judge the likelihood of the newly identified RNAs to engage in biologically relevant interactions with AcnB. However, in the case of *copA*, a regulator of copper homeostasis [65], a functional relationship would be easily deducible. Copper disrupts iron-sulfur cluster assemblies and is toxic for dehydratases like AcnB [66], making a regulatory link between AcnB and *copA* expression plausible. Overall, our data provide further evidence that aconitase might act as an RNA-binding protein not only in eukaryotes, but also in prokaryotes, and propose novel potential RNA targets, suggesting that AcnB engages in regulation beyond its reported autoregulatory feedback loop.Glutamate-5-kinase (ProB)


We considered glutamate-5-kinase a reasonable candidate for moonlighting enzyme function due to its pseudouridine synthase and archaeosine transglycosylase (PUA) domain. PUA domains are principally able to mediate RNA binding [67]. In *E. coli* glutamate-5-kinase, the PUA domain influences catalytic activity and allostery of the adjacent amino acid kinase domain, but is not required for enzymatic activity, rendering its function unclear [39,68]. An RNA-binding function of the domain was previously discussed but remained unconfirmed [69]. The SELEX experiment with glutamate-5-kinase yielded a small number of highly amplified read clusters, located at genomic loci *tktA* (55.2% of reads), *lepB* (30.5% of reads), and *yfjR* (5.8% of reads). We were not able to identify an apparent common feature between transcripts of those genomic loci. However, probing an RNA transcript fragment of *tktA* against *E. coli* glutamate-5-kinase, we observed a stronger complex band relative to a random control RNA fragment in EMSA analysis (Figure 7A). The specificity of the interaction was further confirmed by competition experiments involving a 50-fold excess of unlabeled control RNAs (Figure 7B). Based on this result, we see it as plausible that the PUA domain of *E. coli* glutamate-5-kinase is indeed able to mediate specific RNA binding. We are cautious to consider the RNA fragments identified in our SELEX experiment as primary biological targets of glutamate-5-kinase, due to only three genomic loci being identified with no apparent communalities, and due to a very moderate K_D_ of >>10 µM, as estimated from the concentrations used in our binding assays. However, it is uncertain whether ideal conditions were established for the assays, and at the very least, our results call for further investigations on glutamate-5-kinase as an RNA-binding protein.
Quinone Oxidoreductase (QorA)


*E. coli* quinone oxidoreductase QorA is a structural homologue of the eukaryotic ζ-crystallin, a name referring to the protein’s occurrence in the eye lens of mammals [70]. ζ-crystallin is considered a moonlighting protein. Besides its structural role in the eye lens, it has been shown to reduce *ortho*-quinones [71], but also carries alleged RNA-binding functions, binding to AU-rich elements to regulate mRNAs [42,72,73,74]. While the enzymatic activity of *E. coli* quinone oxidoreductase is not experimentally confirmed, it is suspected that it catalyzes the reduction of large quinone species [75,76]. We used genomic SELEX to scout for the presence of RNA binding in the *E. coli* representative, and the RNA library developed several read clusters, with one read maintaining a striking proportion of 69% of total mapped reads. We generated a sequence logo using the genomic sequences underlying the amplified read clusters as input (Figure 8A). It illustrates a difficult-to-interpret conservation of a 12 nt length nucleotide stretch. We predicted secondary structures of underlying sequences but found no common structural feature. Thus, we simply picked a 27 nt RNA sequence from the top hit (*yffO*), including the motif, and probed for preferential binding in EMSA analysis. The first experiments already presented a strong difference in affinity between the *yffO* mRNA fragment and a random control RNA (Figure 8B).

Due to this clear result, the identified interaction was investigated with more experiments. Further EMSA competition experiments revealed that an 11-fold excess of unlabeled random competitor RNA is not able to impair the QorA-*yffO* mRNA fragment complex band, while 11-fold excess of unlabeled *yffO* mRNA fragment does (Figure 8C), providing solid evidence for specific binding and underpinning the high amplification of the RNA sequence during our SELEX experiment. We subsequently quantified the interaction using EMSA and microscale thermophoresis (MST) titrations as complementary approaches (Figure 8D,E). Fitting of the appropriate binding equations to the titration points delivered concordant K_D_-values of ~7 µM. However, competition experiments with the enzyme’s cofactor NADPH revealed that the cofactor site is likely to participate in RNA binding (Figure 8B,E). NADPH is deeply buried in the enzyme according to a crystal structure [75]. We conducted NMR shift experiments that revealed larger peak shifts for the QorA-NADPH complex compared to the QorA-RNA complex, possibly indicating that the RNA binds only to the surface. QorA contains a Rossmann-fold domain, which has been suspected to be a mediator of RNA-binding [16]. Our results provide further evidence that the Rossmann-fold is a potential RNA-binding site conserved across domains of life. Binding disruption by NADPH was previously reported for the structurally homologous ζ-crystallin in complex with AU-rich RNA strands [72].

The gene *yffO* is part of the grounded CPZ-55 prophage operon [77,78,79,80]. The operon originates from a phage but is permanently integrated into the *E. coli* host genome and thus likely maintains beneficial functions for the host (considering that metabolic burden would lead to deletion of redundant genes in the course of evolution). Assuming biological relevance for QorA-*yffO* mRNA binding, it would comprise an interesting, non-conventional moonlighting interaction, considering the exogenous origin of *yffO*. The exact function of *yffO* is unknown. However, a sequence analysis revealed that the gene product is homologous to the small terminase subunit Gp1 from *Shigella* phage Sf6 (25% sequence identity and 34% similarity in a global sequence alignment; see Appendix A for a structural comparison). The terminase of bacteriophage Sf6 packs newly synthesized concatemeric phage DNA into capsid precursors during virus morphogenesis, and the small subunit Gp1 is responsible for the specific recognition of the concatemeric DNA via a helix-turn-helix motif [81,82]. The structural homology of helix-turn-helix motifs of YffO and Gp1, as well as an N-terminal AT-hook motif [83] with the core sequence GRP (residues 24–26; both marked by a green bar in Appendix A), provide strong indication that YffO has retained its DNA-binding properties. Notably, the surrounding of the AT-hook motif of YffO is more enriched in basic residues than that of Gp1 (PKKRGRPAK vs. EPKAGRPSD). While YffO might have historically served as a small terminase subunit for phage morphogenesis, the protein has by implication lost its function during the transformation of CPZ-55 bacteriophage into a permanently grounded prophage. We suspect that it might have adopted a novel DNA-binding function within *E. coli*. *YffO* is presumably under the control of a σ38-subunit containing polymerase [84,85], indicating a potential involvement of *yffO* in oxidative stress or nutrition starvation [86], which in turn provides a link to the quinone oxidoreductase activity of QorA.

## 3. Discussion

Recent high-throughput studies uncovered dozens of novel putative RNA binding interactions with metabolic enzymes in bacteria [32,33,34] and suggest that up to a quarter of all *E. coli* proteins might bind to RNA. On the other hand, research beside high-throughput screenings is scarce, and further characterization of hypothesized RNA—enzyme interactions is required. Inspired by this shortfall, we set out to validate the frequency of such predicted interactions in protein-centered experiments. As complementary approaches, we performed iCLIP and SELEX experiments in parallel, and tested 19 carefully selected enzymes for their RNA binding properties. EMSA and other biochemical and biophysical methods were used to validate interesting bona fide interactions. In our hands, SELEX turned out to be the more effective method. This entailed that we performed the full preparative iCLIP protocol for only 6/19 candidates that bound to RNA in cells after UV-crosslinking in initial analytical CLIP experiments, but SELEX for 18/19 candidates. The enzymes tested with both methods showed no specificity in RNA binding in either approach. In total, 7/19 candidates showed a detectable RNA binding, two of them with a remarkable sequence specificity. Additionally, we could obtain novel data for the RNA-binding MS2 phage coat protein, which we used as positive control. Table 2 summarizes the detected RNA binding features for these eight proteins.

For 17/19 candidates, we could not detect RNA binding (12/19), or only a rather unspecific binding (5/19), although all selected candidates were high-scoring in at least one of the three high-throughput studies that inspired this work (Appendix A). This raises the question of how reliable the results of high-throughput experiments are. A recent study conducted by Vaishali et al. [87] also touched upon this issue by characterizing the RNA binding properties of six eukaryotic enzymes suggested to be RBPs based on reported mRNA interactome capture. It used NMR spectroscopy and biochemical assays to conclude that binding was either absent or nonspecific, and advised caution when embarking on RNA binding studies involving “unconventional, novel RBPs”. On the other hand, most protein-centered methods like NMR, SELEX or EMSAs study the proteins in their isolated state in vitro, and it should be noted that they might miss moonlighting RNA binding that requires a specific in vivo environment. Summarized, the results by Vaishali et al. and our report highlight the need for more protein-centered investigations.

Five of the 19 candidates (PykF, Pgk, GapA, ThyA, AcnB) only displayed a rather unspecific RNA binding. Because it must be assumed that SELEX will enrich whatever sequence exerts the highest affinity toward the employed protein, the observed selection behavior cannot be interpreted as an unconditional indicator for in vivo-occurring RNA interactions of high affinity and biological significance. For example, the enrichment of A-rich loops observed for Pgk could opportunistically be explained by a coincidental interaction with the ATP-groove with no further biological implications, factoring in the general low affinity of Pgk toward RNAs (estimated K_d_ = 42 μM; Appendix A) and lack of preferential binding compared to a control RNA observed in EMSAs. On the other hand, the interactions might be dependent on other factors that are only available in vivo. It is, therefore, advisable that the interactions that were observed in the form of RNA enrichment during genomic SELEX but were undetectable in follow-up analysis be judged very carefully.

Our experiments with the MS2 phage coat protein not only validated the SELEX procedure, but also provided unanticipated new insights into the features of this protein. First, we could verify a previously reported higher affinity toward hairpin stem-loops with a slightly differing loop sequence (ANCA) compared to the native phage sequence ANUA [54,56]. Furthermore, due to NGS providing higher sequencing power, the RNAs identified by us greatly extend the pre-existing list of putative heterologous RNA targets. The underlying gene functions of respective RNAs overwhelmingly relate to cell surface function, as previously reported [54], but also comprise genes of unknown function. By implication, the enrichment of corresponding mRNAs/asRNAs might give a hint that some of these genes of unknown function are involved in cell surface physiology in one way or another. While the phage mechanism of host-translational manipulation to promote self-proliferation is well-understood [88], implications of a possible specific posttranscriptional control of host cell surface genes are not described in the literature and comprise an intriguing matter to be explored.

SELEX with glutamate-5-Kinase (ProB) revealed a small number of highly amplified RNA fragments. Representative EMSA experiments with one of those sequences confirmed the specific binding. While we cannot pinpoint the site of binding, this result provokes further focus on the notion that the PUA domain of ProB might engage in RNA-binding functions. Notably though, the K_d_ for the interaction was found to be quite weak. A continuation of the experimental validation of these interactions under optimized binding conditions is, therefore, required. Co-purification of ProA, which interacts with ProB and stabilizes it [69], might facilitate RNA-binding experiments. The heterocomplex formation might not only be useful for ProB stability but also be worth examining for altered affinity toward the specific interaction with the RNA fragments.

The most promising interaction we found was that between quinone oxidoreductase (QorA) and the *yffO* mRNA, encoded by a gene of a grounded prophage. However, the observed moderate in vitro K_d_ of 7 μM still suggests caution and requires further optimization of the binding conditions and validation with complementary methods. Unfortunately, the physiological role of both interaction partners is still unknown, rendering it too demanding to elucidate the biological function of this interaction within the frame of this work. The function of the suspected former terminase and now grounded gene *yffO* would be of special interest. Another point of interest is the effect of RNA on NADPH binding. Since we have shown that NADPH disrupts the RNA-QorA complex (Figure 8B), QorA activity might be affected vice versa by RNA. Thus far, assaying QorA activity has not been very productive, as its native substrate is not known, and our attempts with 1,4-benzoquinone showed possible residual, but uncertain activity. A quinone oxidoreductase from *Staphylococcus aureus*, which closely resembles *E. coli* QorA, is reported to catalyze the reduction of 9,10-phenanthrenequinone [89]. The size of the substrate also fits the substrate groove of *E. coli* QorA based on crystal structure analysis [75,76], rendering it a good candidate to unravel the substrate spectrum of QorA. It might also be interesting to test the RNA-binding ability of YhdH, another *E. coli* quinone oxidase that is structurally homologous to QorA and also binds NADPH via a Rossmann fold domain [76]. Additionally, QorA binding to the other sequences we found to be enriched (Figure 8A) and should be evaluated.

In summary, our results indicate that a moonlighting RNA binding function of metabolic enzymes is less frequent than the results of high-throughput studies suggest, but does occur. We also assume that regulatory REM networks might be less frequent in prokaryotes than in eukaryotes due to their overall lower complexity, but our results cannot provide a solid statistical basis for this hypothesis. Our findings underline that careful and detailed experimental validation of high-throughput results is required. We want to point out, however, that our aim was rather to perform initial tests for a representative set of enzymes from different metabolic pathways than to study each putative interaction in detail. In vitro experiments might underestimate or fail to detect native RNA–protein interactions due to missing components or inappropriate binding conditions, and some of the ambiguous interactions we found might be substantiated by improving assay conditions.

## 4. Materials and Methods

### 4.1. Cloning and Protein Purification

Genes encoding RBP candidates were derived from the ASKA library [90], a complete set of N-terminal histidine-tagged ORF clones for *E. coli* K12, with the genes being encoded by IPTG-inducible pCA24N expression plasmids. *ansB* and *proB* were cloned into pET21a, because a C-terminal His-tag gave better yields. The gene for the bacteriophage MS2 coat protein was obtained as GeneArt^TM^ Strings^TM^ DNA fragments from Thermo Fisher Scientific (Waltham, MA, USA) and cloned into pUR22 [91].

Target protein-encoding genes were expressed in E. coli BL21-Gold (DE3) cells. After chemical transformation with the respective plasmid, cells were grown to an OD600 of ~0.7. Gene expression was induced by 0.5 mM IPTG and proceeded over night at 25 °C. Cells were harvested by centrifugation, and pellets were resuspended in purification buffer (50 mM Tris, 150 mM NaCl) containing 10 mM imidazole. For PykF, HEPES was used instead of Tris, because Tris inhibited its activity. Cell disruption was achieved by sonication on ice. Cell debris was removed by centrifugation, and recombinant proteins were purified from the soluble fraction by immobilized metal ion affinity chromatography over HisTrap crude FF 5 mL columns coupled to an Äkta micro device (Cytiva, Marlborough, MA, USA, using purification buffer with increasing imidazole content (up to 750 mM) for elution. For further purification and removal of potential trace Hfq impurities, the proteins were further subjected to preparative size exclusion chromatography on a HiLoad 10/300GL Superdex 75 or 200 column coupled to an Äkta prime system (Cytiva), using purification buffer. The proteins were shock-frozen in liquid nitrogen and stored at −80 °C. Concentrations were determined by absorbance spectroscopy using molar extinction coefficients as calculated by ProtParam [92]. Before usage, proteins were routinely thawed quickly, centrifuged, and continuously kept on ice.

### 4.2. iCLIP

To allow immunoprecipitation, *E. coli* strains carrying an N-terminal (PykF, Upp) or C-terminal (all others) genomic double FLAG-tag modification at the respective gene of interest were constructed in strain DY329 using λ-red mediated recombination [49,50] using the kanamycin resistance cassette, which was not removed subsequently. The modified genes were transferred to strain BW25113 using P1 transduction [93].

For iCLIP, the BW25113 strain*s* were grown in 5 mL selective LB-medium overnight. These pre-cultures were used to uniformly inoculate 1000 mL selective LB-medium, and cultures were shaken at 37 °C and 150 rpm until an OD of 1.6 was reached. Cells were harvested by centrifugation (10 min, 4000× *g*, 4 °C), washed thoroughly, resuspended in a small volume of PBS-T buffer (137 mM NaCl, 2.7 mM KCl, 10 mM Na_2_HPO_4_, 0.5% Tween 20, pH 7.4; the liquid should have only little height to promote uniform UV-light exposure), and poured into 15 cm Petri dishes. From this point, cells were continuously kept on ice. Cells were exposed to 3 × 333 mJ/cm^2^ UV light (λ = 254 nm) with intermediary panning, harvested by centrifugation (10 min, 4000× *g*, 4 °C), shock-frozen in liquid nitrogen, and stored at −80 °C until use.

The iCLIP experiments were conducted as described by Buchbender et al. [48] and represent an advanced library preparation, termed “iCLIP2”. The following adaptions were made to account for the bacterial cell: the frozen cell pellet was resuspended in 4 volumes of RIPA buffer (50 mM Tris-Cl pH 7.4, 150 mM NaCl, 5 mM EDTA, 1× HALT protease inhibitor (Thermo Fisher Scientific)). Cells were sonicated on ice for 2 min at 45% pulse, and afterwards, NP-40 was added to 1% and SDS to 0.1% of the final volume. Cell lysate was incubated for 10 min on ice, vortexing every 3 min. Cell debris was removed by centrifugation in a tabletop centrifuge at 21,000× *g* for 10 min at 4 °C. Limited RNase digestion was performed in lysates diluted with the same volume of RQ1 buffer (40 mM Tris/HCl pH 8, 10 mM MgSO_4_, 1 mM CaCl_2_), and 1 mg/mL RNase A (QIAGEN; high RNase condition), or 2 U/µL RNase I (Thermo Fisher Scientific Ambion™) and 0.004 U/µL DNase I (Ambion™; low preparative RNase condition) for 3 min at 37 °C in a water bath. Then, 150 mM NaCl was added, and samples were centrifuged at 21,000× *g* for 10 min at 4 °C.

Immunoprecipitation was performed in lysates using Anti-FLAG M2 Magnetic Beads (Sigma-Aldrich, St. Louis, MO, USA) for 2 h at 4 °C, and precipitated material was washed 4 times with 1 M NaCl wash buffer (50 mM Tris pH 7.4, 1000 mM NaCl, 0.05% Tween 20) and 2 times with PNK buffer (70 mM Tris/HCl pH 7.5, 10 mM MgCl_2_, 0.05% NP-40) to enable subsequent enzymatic reactions on beads. On-bead phosphatase treatment, RNA linker ligation, PNK-mediated phosphorylation using gamma-[^32^P]-ATP, gel electrophoresis, transfer, RNA elution and iCLIP2 library preparation were performed according to Buchbender et al. [48].

### 4.3. Genomic SELEX

The RNA library was derived from the *E. coli* genome by transcription of adapter-ligated, sheared genomic DNA. The genomic DNA library was obtained commercially using a library preparation service from Microsynth AG (Balgach, Switzerland). In brief, the library was prepared by genome shearing, end-repair to create blunt-ends, 5′-phosphorylation and dA-tail addition, followed by ligation of TruSeq^®^-compatible adapters. The lengths of sequences comprising the DNA library were normally distributed in a range between 300 and 600 bp, as verified by automated gel electrophoresis (Bioanalyzer, Agilent Technologies, Santa Clara, CA, USA) after synthesis. Size distribution was additionally checked by standard gel electrophoresis after each SELEX round. In order to minimize unspecific selection effects originating from the adapter sequence, truncated adapter versions were used during SELEX, and the full-length versions were restored prior to sequencing of evolved libraries. The T7 promoter sequence was added 5′ to the truncated adapter so that first nucleotides were identical in all transcripts (“GGG” plus a constant adapter sequence of 20 bases), preventing any transcriptional selection bias during SELEX. The sequences of all oligonucleotides used for SELEX are listed in Appendix A.

General SELEX functionality was verified by both amplification of aptamers from a random library by hen egg white lysozyme [94], and amplification of operator hairpin sequences from a genomic *E. coli* library by MS2 phage coat protein.

For SELEX, transcription with T7 RNA polymerase (Thermo Fisher Scientific) using α-^32^P-ATP was carried out according to the manufacturer’s protocols. RNA was precipitated with 0.5 volume 0.5 M EDTA pH 8.0, 0.5 volume 7.5 M ammonium acetate, and 1.5 volumes isopropanol at −20 °C. RNA was pelletized by centrifugation and washed at least twice using 70% ethanol. RNA was resolubilized in H_2_O. Concentration was determined by scintillation counting, and the success of transcription reactions was intermittently verified by urea PAGE in between selection rounds.

The RNA library was added in tenfold molar excess to the respective target protein and incubated for 30 min. A solution of 50 mM Tris, 80 mM NaCl, 10 mM KCl, 0.8 mM MgCl_2_, and 0.5 mM DTT, pH 7.5 was used as buffer, rudimentarily resembling native *E. coli* conditions. For PykF, HEPES was used instead of Tris. Prior to protein addition, the RNA library was briefly heated to 95 °C and cooled, inducing unfolding and redevelopment of RNA secondary structures. For the initial 2–3 rounds of SELEX, the RNA was subjected to negative selection [95], eliminating filter-binding RNAs.

After incubation, the RNA-protein solution was filtered through a 0.2 µm M24 cellulose filter slice (Cytiva Whatman™), retaining protein–RNA complexes but not free RNAs [96]. The filter was washed three times with buffer. Phenol-chloroform extraction was used to elute RNAs. For this, the filter was soaked in 400 µL 8 M urea and 500 µL ROTI^®^Aqua-P/C/I pH 4.5 (Carl Roth GmbH, Karlsruhe, Germany) and mixed in an Eppendorf Thermomixer 5355 (Eppendorf SE, Hamburg, Germany) for 10 min. The liquid was taken off, and filter extraction was repeated one time with half volume. The unified liquids were centrifuged for phase separation. The aqueous phase was isolated and extracted with chloroform to remove phenol traces. RNA was precipitated at −20 °C by adding 1/10 volume 3 M sodium acetate pH 6.5, 1 volume isopropanol, and 1 µL GlycoBlue™ Coprecipitant (Thermo Fisher Scientific). RNA was pelletized by centrifugation, washed with 70% ethanol, and dissolved in 70 µL H_2_O.

Reverse transcription was carried out with SuperScript IV reverse transcriptase (Thermo Fisher Scientific) according to the manufacturer’s protocol. The resulting DNA was precipitated at −20 °C by adding 1/10 volume 3 M sodium acetate, 1 volume isopropanol, and 1 µL GlycoBlue™ Coprecipitant. DNA was pelletized by centrifugation and dissolved in 70 µL H_2_O. Subsequently, PCR amplification using Q5 polymerase (New England Biolabs, Ipswich, MA, USA) according to the manufacturer’s protocol was used to replenish DNA, providing enough starting material for the subsequent SELEX cycle. PCR amplification was verified by agarose gel electrophoresis.

### 4.4. Sequencing and Processing of Reads

All DNA libraries were sequenced on an Illumina MiSeq device. Libraries from iCLIP were sequenced using an Illumina 50 cycles MiSeq Reagent Kit v2. Libraries from SELEX were sequenced using an Illumina 500 cycles MiSeq Reagent Nano Kit v2.

For SELEX, in the first step of read processing, adapters were removed using Cutadapt version 2.8 [97] with the following parameters: -overlap = 10-minimum-length = 20-discard-untrimmed. Deduplication was performed using samtools v1.9 (samtools markup -r) [98]. Bowtie2 version 2.4.1 [99] was used to map the reads to the genome of *Escherichia coli* K-12 substrain MG1655 with standard parameter settings.

For iCLIP, read processing was conducted according to a published protocol [100]. To account for the prokaryotic genome, Bowtie2 was used for read mapping to the genome of *E. coli* K-12 substrain MG1655 instead of STAR-mapper.

Mapping data (.bam files) and crosslinking-sites (.bed files) were examined in the most current version of the integrative genomics viewer IGV [101].

### 4.5. Cluster Calling in Mapped Genome Data/Evaluation of Enriched RNAs

Peak calling was mainly achieved by thorough manual inspection of mapping data, applicable to the moderate genome size of *E. coli*. This comprised identification of areas with significantly above-average read density, inspection of peak shapes, and comparison between evolved libraries to ensure specificity of peaks (avoiding peak calls for unspecific amplification effects in SELEX samples). Only deduplicated reads were considered for iCLIP reads. For SELEX libraries, mapped reads were evaluated, and the deduplicated dataset was inspected to validate peak calls. Peak calling by PureCLIP [102] was considered in addition to manual inspection for iCLIP data. For motif inquiries, RNA sequences at the apex of coverage within each peak were considered. Motif inquiries were conducted and visualized as sequence logos with the most recent version of MEME [103].

### 4.6. Electrophoretic Mobility Shift Assays

RNA probes were ^32^P-labeled with T4 polynucleotide kinase (New England Biolabs) according to the manufacturer’s protocol and tested for interactions with respective target proteins in native polyacrylamide gel electrophoresis. Components for native gels were 40% acrylamide/bisacrylamide 18:1 (final concentration 7.5% or 10%), 2.5% *v/v* glycerol, H_2_O, and 1× TB or 0.5× TBE buffer, polymerized by TEMED and APS. For development of secondary RNA structures, all RNAs were heated in assay buffer (see below) for 1 min to 95 °C and subsequently cooled by placing them into a metal rack at room temperature. Pre-incubation of RBP candidates and RNAs started by mixing protein and RNA and proceeded for 30 min at 37 °C under the same buffer conditions used for incubation of the respective RBP candidate and RNA library during genomic SELEX: 50 mM Tris, 80 mM NaCl, 10 mM KCl, 0.8 mM MgCl_2_, 0.5 mM DTT, pH 7.5. For PykF, HEPES was used instead of Tris. In the case of competition assays, target RNA and competitor RNA were treated separately to allow secondary structure formation and were added to the protein concurrently. Samples to be subjected to electrophoresis were supplemented with 10% glycerol to facilitate sinking into gel pockets. For visualization of ^32^P-labeled RNAs, gels were wrapped into plastic foil and exposed to a PhosphoImager screen for at least 2 h. The phosphoscreen was visualized on a Cyclone Storage Phosphor System (Canberra Packard, Meriden, CO, USA). Quantification of bands was performed with the OptiQuant 3.0 Software supplied by Canberra Packard. Data of EMSA titrations were fitted in SigmaPlot 14 using the common hyperbolic binding equation to determine a K_d_-value.

### 4.7. MST Titration

MST assays for quinone oxidoreductase were conducted on a Monolith NT.115Pico device (NanoTemper, Munich, Germany). The Pico RED detector (excitation wavelength 600–650 nm) was used to measure fluorescent Cy5-labeled RNA. Prior to titration experiments, base fluorescence intensities of free RNA and RNA–protein complex solutions were measured to ensure homogeneity of the signal. The buffer used during RNA identification in SELEX was used as assay buffer, supplemented with 0.05% Tween-20 to avoid adsorption of biomolecules to the capillaries. A sigmoidal equation as provided in the NanoTemper evaluation software was fitted to the titration data to obtain K_d_ values.

### 4.8. Secondary Structure Prediction

For prediction of RNA secondary structures, the most recent version of RNAfold [104] was routinely consulted. For critical assessments, additional prediction suites were used in comparative fashion. These included CentroidFold [105], IPKnots [106], RintD [107], and RintW [108].

## Figures and Tables

**Figure 1 ijms-24-11536-f001:**
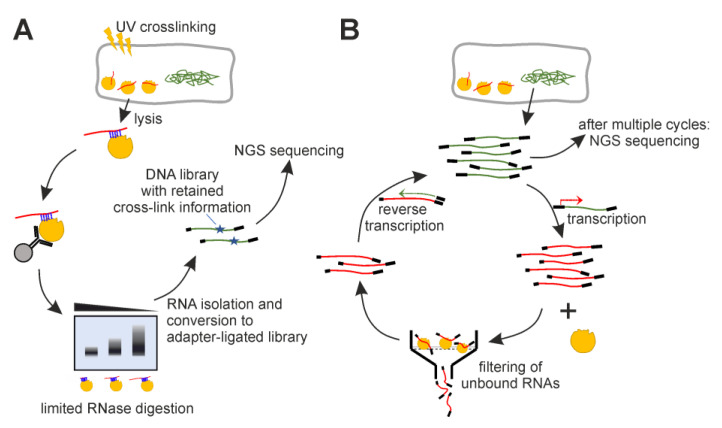
Schematic overview of experiments. (**A**) iCLIP—UV-light exposure induces covalent cross-linking of proteins and RNAs in proximity. Immunoprecipitation isolates the enzyme of interest. If RNA can be detected in a limited RNase titration, it is isolated and processed into an NGS-applicable DNA library by reverse transcription and adapter ligation. (**B**) SELEX—Fragmented *E. coli* DNA is processed into an RNA library by adapter ligation and transcription. The RNA library is exposed to the enzyme, and resulting complexes are isolated via filter binding. The bound RNAs are isolated and reverse-transcribed into DNA to enable PCR re-amplification. The process is repeated multiple times, ideally until the filter binding assay retains significantly increased RNA amounts.

**Figure 2 ijms-24-11536-f002:**
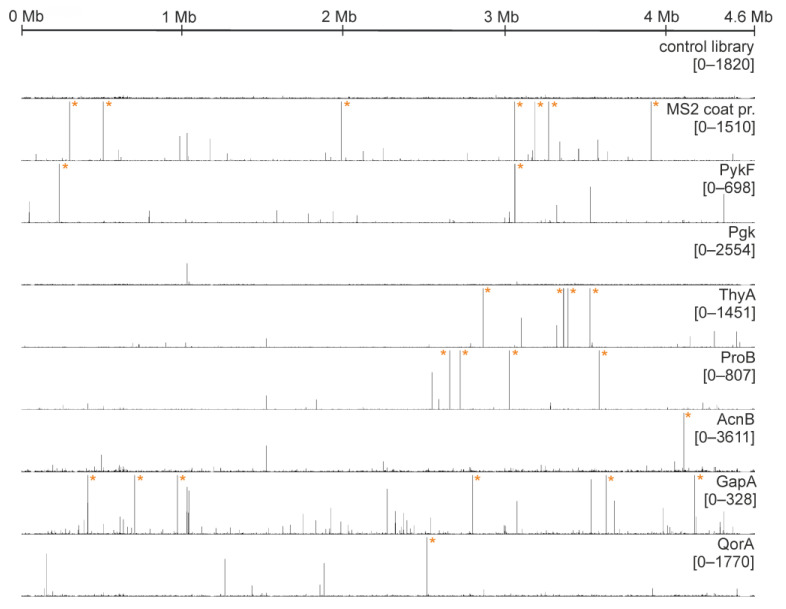
Distribution maps of sequenced reads. The X-axis represents the whole genome. The Y-axis is scaled to 2% of respective total read number for each lane. The numbers below the protein name indicate this 2% range. Asterisks mark read numbers surpassing the displayed scale.

**Figure 3 ijms-24-11536-f003:**
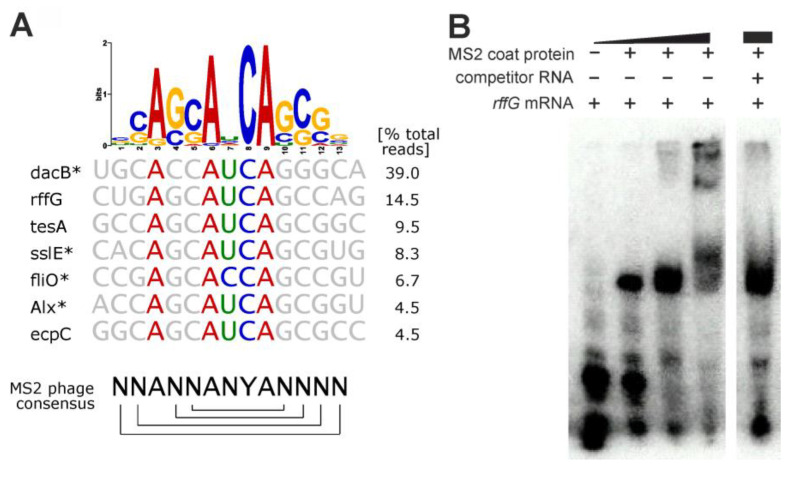
MS2 coat protein SELEX results. (**A**) Sequence logo derived from all genomic sequences mapped to significant read clusters. Below, gene names and aligned sequences are shown for the seven largest read clusters, comprising an accumulated 87% of total mapped reads. On the bottom, the consensus sequence of the MS2 operator hairpin (brackets indicate base pairing) is shown for reference. Asterisks indicate antisense orientation of the enriched RNA relative to the respective gene. (**B**) Competition EMSA validating the interaction between MS2 coat protein (0.34–34 µM) and a radiolabeled 36 nt *rffG* mRNA fragment (700 nM). The competitor RNA (70 µM) is an artificially designed 40 nt fragment with random sequence (all sequences listed in Appendix A).

**Figure 4 ijms-24-11536-f004:**
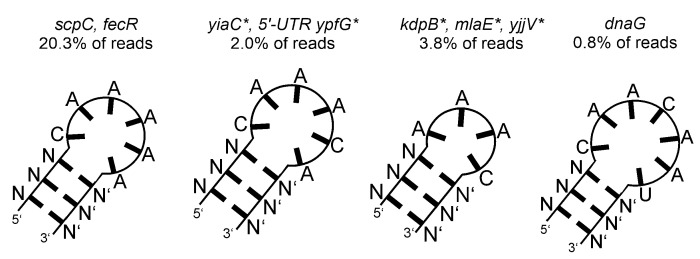
Secondary structure predictions of the top four RNA fragments enriched in the Pgk SELEX experiment. The underlying gene names and the read cluster size (as % of total mapped reads) are indicated. Asterisks indicate antisense orientation of the enriched RNA relative to the respective gene.

**Figure 5 ijms-24-11536-f005:**
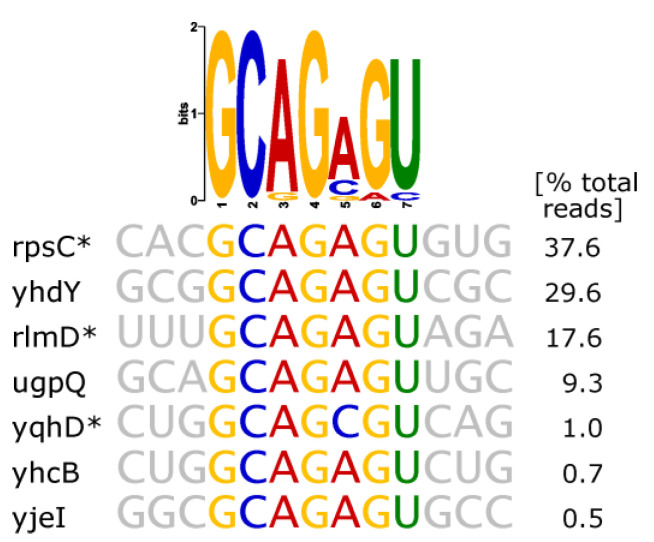
Thymidylate synthase SELEX results. The sequences of largest read clusters and a derived sequence logo are shown. Asterisks indicate antisense orientation of the enriched RNA relative to the respective gene.

**Figure 6 ijms-24-11536-f006:**
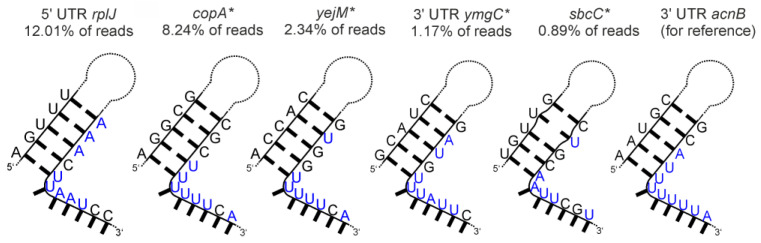
Secondary structure predictions of six RNA fragments enriched in the AcnB SELEX experiment. The underlying gene names and the read cluster size (as % of total mapped reads) are indicated. Asterisks indicate antisense orientation of the enriched RNA relative to the respective gene. The AU-rich sequences are highlighted in blue.

**Figure 7 ijms-24-11536-f007:**
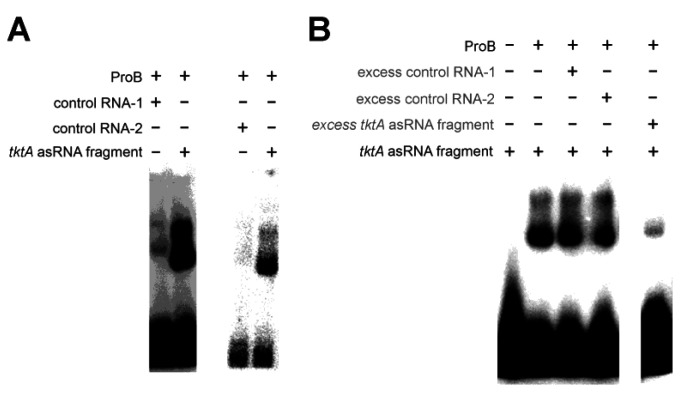
EMSAs validating the specific interaction between glutamate-5-kinase (ProB) and the *tktA* antisense RNA (asRNA) fragment. (**A**) ProB (8 µM), control RNA-1 (2 µM), control RNA-2 (1 µM), *tktA* asRNA fragment (lane 2: 2 µM, lane 4: 1 µM). (**B**) Competition EMSA, unlabeled RNA indicated by grey text. ProB (8 µM), unlabeled control RNA-1 (100 µM), unlabeled control RNA-2 (100 µM), *tktA* asRNA fragment (labeled: 2 µM, unlabeled: 100 µM). RNA sequences are listed in Appendix A.

**Figure 8 ijms-24-11536-f008:**
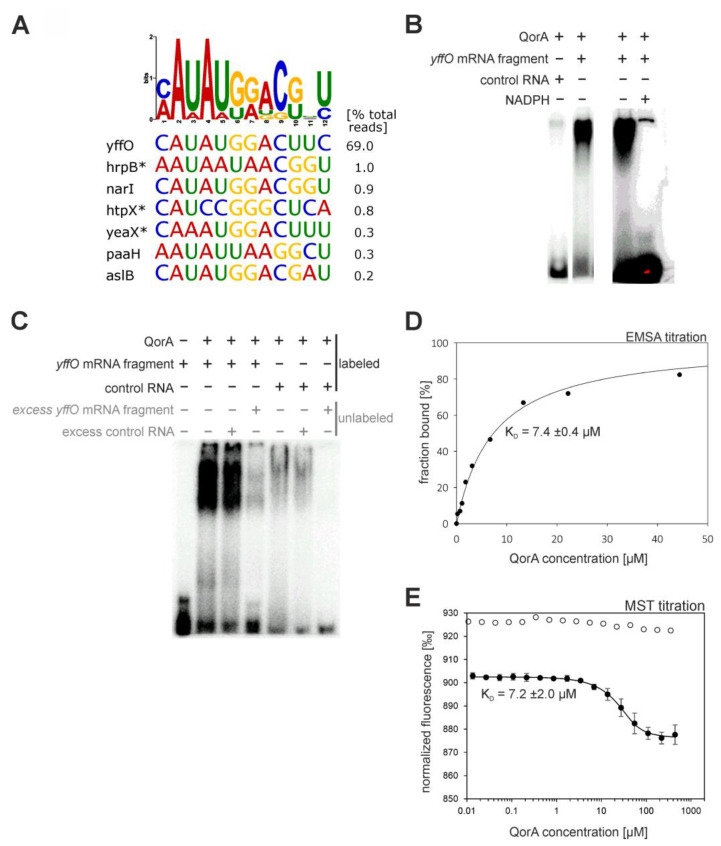
Results and validation of the quinone oxidoreductase (QorA) SELEX experiment. (**A**) Sequence logo and underlying sequences of largest read clusters. Asterisks indicate antisense orientation of the enriched RNA relative to the respective gene. (**B**) EMSAs validating the specific interaction between quinone oxidoreductase (QorA) and the *yffO* mRNA fragment. QorA (14 µM), *yffO* mRNA fragment (12 µM), unrelated control RNA (12 µM), NADPH (20 mM). The red color indicates oversaturation of the picture in that area. (**C**) Competition EMSA, excess unlabeled RNA indicated as grey text. QorA (14 µM), *yffO* mRNA fragment (labeled: 12 µM, unlabeled: 130 µM) control RNA (labeled: 12 µM, unlabeled: 130 µM). RNA sequences are listed in Appendix A. (**D**) Evaluation of EMSA titration of 350 nM *yffO* mRNA fragment. The graph of the fitted hyperbolic binding equation and resulting K_D_ are displayed. (**E**) Evaluation of MST titration of 10 nM Cy5-labeled *yffO* mRNA fragment, followed by relative fluorescence at 670 nm. The graph of the fitted binding equation and resulting K_D_ are displayed. White data points represent an analogous titration in the presence of 10 mM NADPH.

**Table 1 ijms-24-11536-t001:** Selected Enzymes.

Metabolic Pathway	Enzyme	Literature References ^a^
Glycolysis	Pyruvate kinase (PykF)	-
	Phosphoglycerate kinase (Pgk)	binds coding mRNA regions (euk) [31]
	Glyceraldehyde-3-phosphate dehydrogenase (GapA)	binds AU-rich elements (euk) [16,17,18,19]
	Enolase (Eno)	part of RNA degradosome (prok) [35]
	Phosphoglycerate mutase (GpmA)	-
Pentose phosphate pathway	Transaldolase A (TalA)	-
	Transaldolase B (TalB)	-
	Ribose-5-phosphate isomerase (RpiA)	-
Lipopolysaccharides	2-dehydro-3-deoxyphosphooctonate aldolase (KdsA)	-
Nucleotide metabolism	Uracil phosphoribosyltransferase (Upp)	-
	Adenylate kinase (Adk)	general RNA co-purification (euk) [36,37]
	Thymidylate synthase (ThyA)	binds thyA mRNA (prok) [38]
Amino acid metabolism	L-asparaginase (AnsB)	-
	Glutamate-5-kinase (ProB)	Features a PUA domain (prok) [39]
Citric acid cycle	Malate dehydrogenase (Mdh)	binds 3′ UTRs of mRNA (euk) [40]
	Aconitase (AcnB)	binds acnB mRNA (prok) [41]
Oxidoreductases	Superoxide dismutase (SodA)	-
	Quinone oxidoreductase (QorA)	binds pH-responsive elements (euk) [42]
	Pyridoxine/pyridoxamine-5-phosphate oxidase (PdxH)	-
Control	Bacteriophage MS2 coat protein (no metabolic enzyme)	binds phage-specific stem loops (prok) [43]

^a^ Cites the most relevant reference on (potential) RNA binding. Data on prokaryotes (prok) are cited where available. Otherwise, data on homologues from eukaryotes (euk) are cited.

**Table 2 ijms-24-11536-t002:** Studied enzymes with detectable RNA binding.

Protein	Description of Enriched RNA Features	Predicted RNA Secondary Structure ^a^
pyruvate kinase (PykF)	various mRNAs/asRNAs with no apparent communal attribute	-
phosphoglycerate kinase (Pgk)	poorly defined hairpins, showing A-rich loops of variable length as common feature	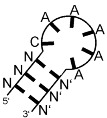
glyceraldehyde-3-phosphate dehydrogenase (GapA)	poorly defined, AU-rich sequence stretches	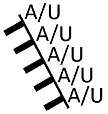
thymidylate synthase (ThyA)	hairpin with strictly conserved sequence GC(AGA)GU (brackets indicating triloop)	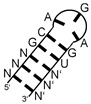
aconitase (AcnB)	poorly defined stem-loops with 3′-adjacent U-rich stretches	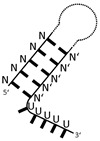
glutamate-5-kinase (ProB)	small number of specific fragments (two mRNAs and one asRNA)	-
quinone oxidoreductase (QorA)	few specific mRNAs/asRNAs with a potential motif (see Figure 8A). top-scoring: *yffO* mRNA fragment.	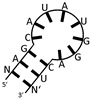
positive control MS2 phage coat protein	bacteriophage operator hairpin-like motif NANN(ANCA)N′N′N′ (N and N′ indicate complementary bases, brackets indicate loop)	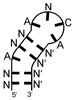

^a^ manually curated structures based on different predictions, see Section 4.

## Data Availability

The raw sequencing data presented in this study are openly available in sequencing_data_Klein_et_al.zip at http://doi.org/10.5283/epub.54333, accessed on 14 July 2023.

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
