# Peer review of "Investigating the Prevalence of RNA-Binding Metabolic Enzymes in E. coli"

_ijms, 2023, doi:10.3390/ijms241411536_

Round 1
Reviewer 1 Report
This manuscript by Babinger and colleagues aimed to investigate the REM hypothesis, which proposes a link between RNAs, metabolic enzymes, and metabolites in cellular regulation. The specific focus of their study was to analyze the binding of RNA molecules to various metabolic enzymes in prokaryotes, which had been previously identified through high-throughput assays. To validate the interaction between RNA and metabolic enzymes, the authors utilized biochemical approaches, such as iCLIP, SELEX, and/or EMSA. These methods validated the RNA: metabolic enzyme interactions, exemplifying their findings with representative enzymes from different metabolic pathways.
Considering the theme of special issues: Small Prokaryotic Proteins Interacting with Nucleic Acids 2.0, this manuscript demonstrated some biochemical evidence to introduce proof-of-principles of REM theory in the E.coli system. However, as noted by the authors, some metabolic proteins do not support the consistent conclusions between two biochemical assays (iCLIP and SELEX). While iCLIP can capture transient or stable Protein: RNA complexes in vivo, in vitro SELEX enriches stable RNAs to a target protein. This inconsistency poses a challenge in fully substantiating the main conclusions of the manuscript, potentially undermining its overall persuasiveness. Therefore, considering the limitations and challenges in establishing consistent findings, it may be advisable to consider submitting this manuscript to an alternative publication venue.
Reviewer 2 Report
The current study aims to predict RNA binding interaction for metabolic enzymes in bacteria.
The author have used interesting techniques like iCLIP, SELEX and EMSA to validate the frequency of protein interaction in protein centered experiments.
The author screened 19 metabolic enzyme out of which only 7 candidate show promising interaction with RNA.
The current study is a preliminary work to identify RNA binding motifs, further studies need to be done.
The results are interesting but all the work done is invitro, for a better assement of the dat a invivo experiments need to be done.
Reviewer 3 Report
This manuscript, by Klein et al, reported a protein-centered high-throughput investigation on RNA binding metabolic enzymes in E.coli. Nineteen E.coli enzymes were selected based on previous high-throughput analysis, and the specific RNA interactions were investigated by in vitro SELEX. Most of the enzymes did not show specific binding, whereas ProB and QorA displayed novel and specific binding to certain RNA motifs. The specific interaction was also validated by competition EMSA or MST. Overall, this manuscript was written with sufficient details provided, appropriate experimental controls were considered during the experimental design, and the results were adequately discussed.
Minor comments are:
1. I wonder if more details can be provided in terms of the DNA library used in SELEX, such as the length of genomic DNA fragments?
2. The T7 polymerase transcription efficiency often correlates with the identities of the first several nucleotides. Could the authors comment on whether bias was introduced during the in vitro transcription step in the SELEX experiment?
3. Section 4.6, EMSA: More details should be given in terms of how RNAs were folded, the incubation time and temperature for the protein and RNA interactions, and how the competitor RNAs were added (i.e., added after target RNA-protein incubation or added concurrently)
Round 2
Reviewer 1 Report
No. My concerns have been addressed by the authors.Reviewer 2 Report
The authors have provided sufficient information and modified the manuscript to be published.